# Effects of a Multidisciplinary Residential Nutritional Rehabilitation Program in Head and Neck Cancer Survivors—Results from the NUTRI-HAB Randomized Controlled Trial

**DOI:** 10.3390/nu12072117

**Published:** 2020-07-17

**Authors:** Marianne Boll Kristensen, Irene Wessel, Anne Marie Beck, Karin B. Dieperink, Tina Broby Mikkelsen, Jens-Jakob Kjer Møller, Ann-Dorthe Zwisler

**Affiliations:** 1Department of Nursing and Nutrition, University College Copenhagen, Sigurdsgade 26, 2200 Copenhagen N, Denmark; ambe@kp.dk; 2REHPA, The Danish Knowledge Centre for Rehabilitation and Palliative Care, Odense University Hospital, and Department of Clinical Research, University of Southern Denmark, Vestergade 17, 5800 Nyborg, Denmark; Tina.Broby.Mikkelsen@rsyd.dk (T.B.M.); Jens-Jakob.Kjer.Moller@rsyd.dk (J.-J.K.M.); Ann.Dorthe.Olsen.Zwisler@rsyd.dk (A.-D.Z.); 3OPEN, Odense Patient data Explorative Network, Odense University Hospital, J.B. Winsløws Vej 9A, 5000 Odense C, Denmark; 4Department of Otorhinolaryngology, Head and Neck Surgery & Audiology, Rigshospitalet, Blegdamsvej 9, 2100 Copenhagen Ø, Denmark; Irene.Wessel.01@regionh.dk; 5Dietetics and Clinical Nutrition Research Unit, Herlev and Gentofte Hospital, Borgmester Ib Juuls Vej 50, 4. 2730 Herlev, Denmark; 6Research Unit of Oncology, Department of Oncology, Odense University Hospital, Sdr. Boulevard 29, 5000 Odense C, Denmark; Karin.Dieperink@rsyd.dk; 7Department of Clinical Research, University of Southern Denmark, J.B. Winsløws Vej 19.3, DK-5000 Odense C, Denmark

**Keywords:** head and neck cancer, rehabilitation, survivorship, eating problems, late effects, quality of life

## Abstract

Head and neck cancer survivors frequently experience nutritional challenges, and proper rehabilitation should be offered. The trial objective was to test the effect of a multidisciplinary residential nutritional rehabilitation programme addressing physical, psychological, and social aspects of eating problems after treatment. In a randomized controlled trial, 71 head and neck cancer survivors recruited through a nationwide survey were randomized to the program or a wait-list control group. Inclusion was based on self-reported interest in participation. The primary outcome was change in body weight. Secondary outcomes included physical function, quality of life, and symptoms of anxiety and depression. Differences between groups at the 3-month follow-up were tested. No significant differences were seen in body weight change, but there were overall trends towards greater improvements in physical function (hand grip strength: *p* = 0.042; maximal mouth opening: *p* = 0.072) and quality of life (“Role functioning”: *p* = 0.041; “Speech problems”: *p* = 0.040; “Pain”: *p* = 0.048) in the intervention group. To conclude, a multidisciplinary residential nutritional rehabilitation program had no effect on body weight in head and neck cancer survivors with self-reported interest in participation, but it may have effect on physical function and quality of life. Further research on relevant outcomes, inclusion criteria, and the program’s effect in different subgroups is needed.

## 1. Introduction

In 2018, approximately 900,000 individuals worldwide were diagnosed with head and neck cancer (HNC) [1]. The incidence has increased in recent years [1,2] with a simultaneous increase in the relative survival [3]; however, despite the prospect of a successful curative treatment result, many HNC survivors feel unprepared for the life that awaits them after treatment [4,5,6,7]. Nutrition impact symptoms such as dysphagia, xerostomia, trismus, and dysgeusia are frequent [8,9,10] and may persist years after treatment [8,9,10,11,12]. These symptoms lead to eating problems, which have substantial negative consequences for HNC survivors’ nutritional status, quality of life (QOL), and daily lives [4,5,10,13,14,15,16,17,18]. Group-based residential rehabilitation programs, where the daily meals are part of the intervention, can provide a safe environment for HNC survivors to practice eating skills [4,19]. Hence, they may be particularly effective to support HNC survivors in the trial-and-error approach; a frequently used coping strategy with continuous experiments to find tolerated foods as the eating problems vary over time [4,6,20,21]; this is a process that may otherwise be complicated by fear of choking [4,22,23] and feelings of defeat associated with unsuccessful experiments [4,20].

To our knowledge, very few studies have explored the potential of group-based residential rehabilitation programs in HNC survivors. In a pilot study testing a 1-week residential psychoeducational program in HNC survivors, high participant satisfaction and improvements in QOL scales were reported [19]. In another pilot study conducted by the researchers behind the present trial, qualitative data showed that HNC survivors benefitted from participating in a multidisciplinary residential nutritional rehabilitation program [4], and significant improvements in body weight and several QOL scales were seen at the 3-month follow-up [24]. With no control group in any of the pilot studies, the effect of residential rehabilitation programs in HNC survivors should be tested in randomized controlled trials. Thus, we designed the NUTRI-HAB trial [24]. The primary objective of the trial was to test the effect of a multidisciplinary residential nutritional rehabilitation program compared to standard care on the primary outcome body weight and secondary outcomes physical function, health-related QOL, and symptoms of anxiety and depression in HNC survivors.

Secondary exploratory objectives and analyses were further predefined in the trial protocol [24]. These will be approached in future publications.

## 2. Materials and Methods

### 2.1. Trial Design

A randomized controlled trial was carried out from May 2019 to December 2019. Participants were randomized to either an intervention group participating in a multidisciplinary residential nutritional rehabilitation program from baseline to 3-month follow-up or a wait-list control group. For the primary objective, data were collected at baseline and 3-month follow-up (Figure 1). Further data collected for explorative objectives will be presented in future publications.

The detailed trial protocol [24] was developed in accordance with the standard protocol items for randomized trials (SPIRIT) 2013 [25,26] statement, the consolidated standards of reporting trials (CONSORT) extension for reporting trials of nonpharmacologic treatments [27], and the template for intervention description and replication (TIDieR) [28] checklist and guide. An overview of trial materials and information on how to obtain these have been published with the trial protocol [24]. The CONSORT 2010 statement [29] and the CONSORT extension for reporting trials of nonpharmacologic treatments [27] were used as guidelines for reporting trial results.

### 2.2. Participants and Setting

Participants were recruited among respondents of the nationwide cross-sectional NUTRI-HAB survey on nutritional challenges, late effects, and QOL in HNC survivors. The survey population was identified through The Danish Head and Neck Cancer Group’s (DAHANCA) national clinical quality database [30] and included all Danish individuals ≥18 years treated with radiation therapy of curative intent for oral, pharyngeal, or laryngeal cancer 1–5 years before survey distribution (*n* = 1937). Since rehabilitation interventions should be based on the wishes and goals of the individual patient [31], a crucial inclusion criterion in the present trial was the individuals’ self-reported interest in participating in the program. Hence, based on self-reported information collected through the NUTRI-HAB survey, respondents were considered eligible for participation in the present trial if they met the following inclusion criteria: (1) Had no active HNC or other cancer at the time of completion of the survey, (2) were self-reliant (defined as having answered “Not at all” to the question “Do you need help with eating, dressing, washing yourself, or using the toilet?” in The European Organization for Research and Treatment of Cancer’s (EORTC) QLQ-C30 questionnaire [32]), (3) were able to speak and understand Danish, and (4) had confirmed that they were interested in participating in a multidisciplinary residential nutritional rehabilitation program at specific dates and had given their permission to be contacted with further information.

All individuals who had responded within nine weeks from survey distribution and who met the inclusion criteria were randomized and placed in random order on numbered invitation lists for intervention group or wait-list control group in an allocation ratio of 1:1. In the recruitment process, the first individuals on each invitation list received further information about the trial and were invited to participate, and if an individual declined the invitation, the next person on the given invitation list was invited. 

Randomization was performed in STATA/IC 15.1 by a blinded researcher who was not involved in the trial intervention or outcome assessment. Randomization was stratified by need for rehabilitation services measured by the REHPA scale, a numerical scale where participants indicate how close or how far they are from living the life they want after or despite their disease [33]. A score of 1 indicates “Very close” whereas a score of 9 indicates “Infinitely far away”. Randomization was stratified to ensure similar proportions of individuals with a score of ≥3 across invitation lists.

The trial was carried out at REHPA, the Danish Knowledge Centre for Rehabilitation and Palliative Care in Nyborg, Denmark, and the intervention was an additional offer to existing rehabilitation services. In Denmark, cancer treatment and rehabilitation services are funded by government taxes and free of charge for patients, and while rehabilitation during treatment is offered at the hospitals, posttreatment rehabilitation is primarily a municipal responsibility [34]. Denmark has 98 municipalities with great variation between their rehabilitation services [35,36].

### 2.3. Intervention

The trial intervention was a multidisciplinary residential nutritional rehabilitation program with a primary focus on the physical, psychological, and social aspects of eating problems after treatment for HNC. The program comprised five-day initial residential stay and two-days follow-up residential stay after three months and consisted of group-based patient education sessions and a few individual activities. The program is based on REHPA’s and former Rehabilitation Centre Dallund’s core program developed through available evidence and more than 10 years’ experience in offering multidisciplinary residential rehabilitation programs for heterogeneous groups of cancer survivors [33,37,38]. The core model was further developed to meet the specific rehabilitation needs of HNC survivors through available evidence, patient involvement and a pilot study including 40 HNC survivors [4]. The program is described in further details in the trial protocol [24], and a schedule of activities during the residential stays is provided in Table 1.

Sessions specifically aimed at managing eating problems included a group session with a clinical dietitian on dietary advice, individual counselling with a clinical dietitian, a practical kitchen workshop with take-home recipes, a group session on oral hygiene and dental reimbursement rules, and instruction in swallowing exercises by an occupational therapist, who are typically responsible for dysphagia management in Denmark [39]. Participants received an exercise manual and a training diary and were encouraged to continue doing the exercises when they came home. During the residential stays, participants stayed at the premises, and all meals were served there. Foods of different flavors and textures were served to allow participants to experiment and to support their trial-and-error coping process as described in the introduction [4,6,20,21]. Physical activity sessions with physiotherapists included restorative yoga and sessions where participants were introduced to different kinds of physical activity that they could do at home, e.g., balance or resistance training exercises. Exercises were adjusted to the participants’ training level. Other activities included group sessions with a psychologist, a session on motivation and action plans, a group conversation with a priest on existence, massage therapy, and optional sessions on vocational counselling, fatigue, and sexuality and intimacy. Individual counselling sessions with relevant professionals (e.g., a speech pathologist or physician) were scheduled depending on the individual participant’s needs, assessed through patient-reported outcome measures. Between the initial stay and the two-day follow-up, all participants had two telephone consultations with a clinical dietitian scheduled in week 4 and week 8 to follow up on the individual consultation at the residential stay, to answer potential emerging questions, and to encourage the participant to continue with any activities or changes that they planned to implement after the residential stay.

Each scheduled program had a maximum capacity of 20 participants. The program was free of charge for participants.

### 2.4. Wait-List Control Group

From baseline to the 3-month follow-up, the wait-list control group received no intervention other than standard care. In Denmark, HNC patients attend follow-up visits at oncological tertiary centers every 6 months for the first 2 years, and annually for the next 3 years. As needed and on referral, they can participate in the municipal rehabilitation services, which, as described, vary across municipalities. Hence, with participants being from all over the country, standard care could vary, and participants were not restricted from participating in other rehabilitation services during the trial. The wait-list control group was offered participation in the multidisciplinary residential nutritional rehabilitation program at the 3-month follow-up stage.

### 2.5. Data Collection and Outcome Measures

All physical measurements and tests were performed by authorized health professionals following strict protocols [24]. Blinding of health professionals performing the measurements was not possible. For the intervention group, the baseline and 3-month follow-up physical measurements were performed at REHPA in the beginning of their five-day and two-day residential stays. The same was the case for the 3-month measurement in the control group. The baseline physical measurement in the control group was performed in one of three outpatient clinics depending on the participant’s place of residence. Patient-reported outcome measures were assessed through online or paper-based questionnaires distributed to participants one week before the physical tests. Research electronic data capture (REDCap) [40] was used for online questionnaires and data storage. Data from paper-based questionnaires and physical measurements were entered in REDCap by one researcher, and the entered data was doublechecked by another researcher.

To reduce missing data, participants who dropped out of the trial were still encouraged to participate in follow-up measurements. Hence, if participants from the intervention group did not participate in the follow-up residential stay, they were encouraged to participate in physical measurements in the nearest outpatient clinic instead, and the same was the case for individuals in the wait-list control group, if they chose not to participate in residential rehabilitation program at the 3-month follow-up. If they were unable to participate in the physical measurements, they were still encouraged to fill out the questionnaires.

#### 2.5.1. Participant Characteristics at Baseline

Information on age, gender, cancer diagnosis, and time interval since treatment was already obtained from DAHANCA’s national clinical database [30]. Questions on current cancer status and respondents’ participation in other rehabilitation services prior to baseline was included in the NUTRI-HAB survey. At the follow-up, this information was collected in the consultations with the clinical dietitian to allow for sensitivity analyses on effect of potential cancer relapse or participation in other nutritional rehabilitation programs on intervention effect.

Nutritional risk was assessed with nutritional risk screening 2002 (NRS 2002) and the scored patient-generated subjective global assessment short form (PG-SGA SF). In the secondary screening with NRS 2002, the overall score comprises an A-score for nutritional status, a B-score for disease severity, and an extra point if aged 70 or above. A higher score indicates greater nutritional risk [41]. The PG-SGA SF includes questions on weight changes, changes in dietary intake (amount or texture), nutrition impact symptoms, and performance status [42]. The score ranges from 0–36, and a higher score indicates a higher risk of malnutrition. The Danish version has been translated, cross-culturally adapted, and linguistically validated [43] and was used with permission. Further data used in the assessment of nutritional status included body mass index (BMI) and participants’ current body weight (percentage) in relation to their precancer body weight. Questions on precancer body weight and participants’ own evaluation of current body weight were included in the nationwide cross-sectional survey prior to inclusion.

The REHPA scale was included in the baseline questionnaires with patient-reported outcome measures. In addition to the numerical scale, participants could mark the challenges preventing them from achieving their goals. In combination with the other patient-reported outcome measures, this information was used to target the intervention to the individual participant’s rehabilitation needs.

#### 2.5.2. Primary Outcome

The primary endpoint was percentage change in body weight. Body weight was measured to the nearest 0.1 kg using calibrated and leveled Seca 877/878 scales, and participants were instructed to minimize their food and fluid intake two hours before the weighing.

#### 2.5.3. Secondary Outcomes

Secondary outcomes included BMI and changes in measures of physical function, patient-reported outcome measures of health-related QOL, and symptoms of anxiety and depression.

To allow for the estimation of BMI (body weight in kg divided by squared height in meters), height was measured to the nearest 0.5 cm using a Seca 222 stadiometer. Measures of physical function were maximal mouth opening, hand grip strength, the 30-s chair stand test, and 6-min walk test. Maximal mouth opening was measured in mm using a TheraBite^®^ range-of-motion (ROM) scale. Hand grip strength was measured in kg using a calibrated Jamar hydraulic hand dynamometer. All measurements were made with the hand dynamometer in the second handle position, and three consecutive measurements in each hand were performed. The highest of the six measurement was used in the data analyses [44]. The 30-s chair stand test was used to assess lower body strength [45], and the registered score was the number of full stands from a chair during 30 s without using the hands. If participants were unable to rise without using their hands, it was registered that the test was completed in a modified version and the following tests for that participant were completed in the modified version. The 6-min walk test was used to measure the submaximal level of functional capacity [46]. The test was performed on a 30-m walking course, and the score was the total distance walked in meters.

Health-related QOL was measured using the Danish translations of the EuroQol 5D-5L (EQ-5D-5L) [47], the EORTC QLQ-C30 [32,48], and the diagnosis-specific EORTC QLQ-H&N35 [48,49]. The EQ-5D-5L covers mobility, self-care, usual activities, pain/discomfort, and anxiety/depression, and overall health is measured using the visual analogue scale (VAS) and a summary index score based on the five dimensions and on societal preference weights for the health state. The VAS scale ranges from 0–100, and the summary index score calculated based on Danish values ranges from −0.624 to 1.0. A higher score indicates better self-rated health [47]. The EORTC QLQ-C30 comprise one global QOL scale, five functional scales and nine symptom scales, whereas the QLQ-H&N35 comprise 18 symptom scales. All EORTC scales range from 0–100. A higher score indicates a higher response level. Thus, a high score for a functional scale or global QOL indicates a high level of functioning/QOL, and a high score on a symptom scale indicates a high symptom level [32,48,49].

Symptoms of anxiety and depression were measured with the Danish translation of the hospital anxiety and depression scale (HADS). The two subscales for anxiety and depression range from 0–21 points, and a higher score indicates a higher symptom level [50].

### 2.6. Sample Size

Based on results from a previous pilot study [4,24], a sample size of 30 individuals in each group was required to detect a difference of 1.74 ± 2.37 in percentage body weight change with a power of 80% and a significance level of 5%. Hence, with an estimated withdrawal rate of 15% [4], the aim was to include 36 participants in each group.

### 2.7. Statistical Analyses

Descriptive statistics were used to summarize baseline data. The intervention effect on the primary outcome, the percentage change in body weight, was analyzed by both the intention-to-treat principle and per protocol principle, whereas intervention effect on other outcome measures were analyzed by the per protocol principle.

In the intention-to-treat analysis of intervention effect on primary outcome, multiple imputations (m = 20) were used to account for missing data under a missing-at-random assumption [51]. Missing observations in body weight at the 3-month follow-up were imputed using the following variables: Baseline body weight, treatment arm (group), age, gender, cancer diagnosis, time interval posttreatment, and REHPA scale score < 3/≥ 3 at inclusion. In the per protocol analyses, only participants with baseline and follow-up measurements of the given outcome were included.

Development in outcome scores from baseline to 3-month follow-up were calculated for each participant, and differences between the intervention group and wait-list control group were tested. In the intention-to-treat analysis, difference between groups in percentual change from baseline to follow-up were assessed using linear regression. In the per protocol analyses, differences were tested using a two-sample two-sided *t*-test for normally distributed data and Mann–Whitney U test for non-normally distributed data. As described in the trial protocol [24], the effect size for normally distributed data was estimated with Cohen’s d [52]. However, for non-normally distributed data, where differences were tested using the Mann–Whitney U test, it was more appropriate to estimate the effect size (r) by dividing the z value obtained in the Mann–Whitney U test with the square root of the number of observations [53]. It is suggested that Cohen’s d values of 0.8, 0.5, and 0.2 represent large, medium, and small effect sizes, while the corresponding values for r are 0.5, 0.3, and 0.1 [52,53]. 

Adjusted analyses were performed using multiple linear regression to adjust for gender, time interval (months) posttreatment, and REHPA scale score.

In addition to the analyses defined in the trial protocol, differences within groups from baseline to follow-up were tested with a two-sided paired *t*-test for normally distributed data and with the Wilcoxon signed-rank test for non-normally distributed data. For outcomes on physical function where evidence on minimal clinically relevant change was available, it is indicated in the result tables whether mean changes within groups from baseline to follow-up are greater than this cut-off. The relevant cut-offs were defined as 5% for body weight [41], 5 kg for hand grip strength [54], and 14 meters for the 6-min walk test [55].

Participants who had a relapse of their cancer during the trial were not excluded from data analyses, but in accordance with trial protocol [24], sensitivity analyses were performed to assess whether this affected results.

A statistical significance level of 5% was applied. Per protocol analyses of differences between groups were performed in SAS^®^ Enterprise Guide^®^ 7.1 by a blinded researcher, and the project group interpreted results before unblinding. STATA/IC 16.0 was used for other data analyses.

### 2.8. Ethical Statement

The trial was conducted in accordance with the Declaration of Helsinki [56]. Informed written consent was obtained from all participants, and they were informed verbally and in writing that participation was voluntary, and that they could withdraw their consent at any time. The Regional Committees on Health Research Ethics for Southern Denmark assessed the duty to notify for the trial (journal number 20182000-165) and concluded, based on Danish legislation, that the trial was not subject to the duty to notify since no biological material was included. The trial was registered by The Danish Data Protection Agency, registration number 2012-58-0018, approval number 18/14847, and registered in the database, Clinical Trials (www.clinicaltrials.gov, NCT03909256), before inclusion of participants. Furthermore, a detailed trial protocol was published to verify adherence to original intent [24].

## 3. Results

In total, 71 individuals were included, of whom 36 were randomized to the intervention group, and 35 were randomized to the control group. Participant baseline characteristics are shown in Table 2.

In both groups, three participants were lost between baseline and 3-month follow-up (Figure 2).

In the intervention group, an additional six participants did not participate in the two-day follow-up residential stay at the 3-month follow-up, but they still completed the patient-reported outcome questionnaires and/or participated in the physical measurements at the outpatient clinics. These participants still had telephone consultations with the clinical dietitian in weeks 4 and 8. Since outcome measurements for all participants in the intervention group were scheduled in the beginning of the two-day follow-up residential stay, and hence, not measured the effect of the two days, the six participants were categorized as having completed the intervention from baseline to 3-month follow-up and included in per protocol analyses.

Three participants from the intervention group and one participant from the control group had relapses of their cancer. Two of them were among the three participants that were lost to the follow-up in the intervention group, whereas the follow-up data were available for the remaining two.

An overview of the intervention group’s scheduled individual counselling sessions with professionals other than a clinical dietitian during the five-day residential stay and their choices of optional group sessions is provided in Appendix A.

### 3.1. Intervention Effect on Primary Outcome

In the intention-to-treat analysis, the primary endpoint, the percentage change in body weight, was almost identical across groups (0.46% in intervention group vs. 0.38% in control group), and the adjusted *p*-value for differences between groups was 0.795 (Table 3). The per protocol analysis yielded similar results (adjusted *p* = 0.752). No statistically significant or clinically relevant changes within groups were seen.

### 3.2. Intervention Effect on Secondary Outcomes

For changes in maximal hand grip strength, significant differences (*p* = 0.038) were seen between groups with a mean increase of 1.3 kg in the intervention group, while a slight decrease (−0.6 kg) was seen in the control group. With Cohen’s d of 0.55, this corresponded to a medium effect. The differences remained significant in the adjusted analyses (Table 3). In the 30-s chair stand test, improvements were greater in the control group than in the intervention group (2.3 vs. 0.5, d= −0.69, adjusted *p*-value = 0.008). For maximal mouth opening and the 6-min walk test, tendencies were seen towards greater improvements in the intervention group (mouth opening: d = 0.46; *p* = 0.088; 6-min walk test: d = 0.51; *p* = 0.061), but in the adjusted analyses the tendency was no longer present for the 6-min walk test. In the intervention group, a statistically significant and clinically relevant improvement in the 6-min walk test (*p* < 0.001) was seen from baseline to follow-up (Appendix B, Table A2). In the control group, the result of the 30-s chair stand test improved significantly (*p* < 0.001).

No significant differences were seen between groups or within groups in EQ-5D-5L scores (Table 4).

In the EORTC QLQ-C30 scales, significant differences were seen between groups in “Role functioning” (r = 0.28; *p* = 0.024, adjusted *p* = 0.041) and “Pain” (r = −0.27, *p* = 0.029, adjusted *p* = 0.048), indicating greater improvements in the intervention group. The same was the case for “Fatigue” (r = −0.24, *p* = 0.050), but in the adjusted analysis, only a tendency was seen (Table 4). For “Physical functioning”, a tendency towards greater improvement in the intervention group was seen (r = 0.22, *p* = 0.070), but in the adjusted analysis this tendency was no longer present. In the intervention group, a significant improvement in “Cognitive functioning” was seen from baseline to follow-up (*p* = 0.034), while no significant changes were seen in the control group.

In the EORTC QLQ-H&N35 scales, improvements in “Speech problems” were greater in the intervention group (r = −0.18, adjusted *p* = 0.040), but so was the increase in the “Felt ill” symptom level (r = 0.29, adjusted *p* = 0.020) and the use of “Nutritional supplements” (r = 0.31, adjusted *p* = 0.005). From baseline to follow-up, the intervention group had significant improvements in the symptom scales “Swallowing” (*p* = 0.032), “Speech problems” (*p* = 0.009), “Trouble with social eating” (*p* = 0.027), “Teeth” (*p* = 0.023), “Opening mouth” (*p* = 0.020), and “Dry mouth” (*p* = 0.029). From baseline to follow-up, the control group had significant decreases in symptom levels in “Swallowing” (*p* = 0.010), “Senses problems” (*p* = 0.007), “Coughing” (*p* = 0.038), and “Nutritional supplements” (*p* = 0.025) (Table 4).

For HADS scores, a tendency towards greater improvements in the anxiety subscale was seen for the intervention group (adjusted *p* = 0.061), but no significant differences were seen between groups or within groups (Table 4).

### 3.3. Sensitivity Analyses

Sensitivity analyses of differences between groups when excluding participants with relapses of their cancer did not change the overall results (Appendix B, Table A3). Since none of the participants participated in other nutritional rehabilitation services during the trial period, no sensitivity analyses were required to account for this.

## 4. Discussion

This is the first randomized controlled trial to test the effect of a multidisciplinary residential nutritional rehabilitation program compared to standard care in HNC survivors. The trial showed no effect on the primary outcome, the percentage change in body weight, but there was an overall trend towards greater improvements in physical function and QOL in the intervention group.

In the pilot study of the intervention, a significant increase in body weight was seen [4,24], and several factors may contribute to why no effect on body weight was seen in the present trial. Since participants were 1–5 years posttreatment, it can be questioned whether change in body weight was the most relevant primary outcome. It was chosen because it is an objective measure, and in the nationwide survey that preceded participant recruitment, approximately half of HNC survivors 1–5 years posttreatment had not regained their habitual weight. This was also the case for trial participants; however, while 51% had a current body weight lower than 95% of their precancer weight, only 11% considered their current weight too low, and 46% considered it too high (Table 2 and Appendix C, Table A4). Hence, for most participants, increase in body weight was not a desired outcome. The individual counselling sessions with the clinical dietitian were tailored to the individual participant, which meant that for some participants, it included strategies for weight gain while for others, it included strategies for weight loss. Hence, no overall effect on body weight was seen. Furthermore, body weight is only a crude measure of nutritional status. Information on body composition could have been relevant, since body composition may change even though the weight is stable. Subjective assessment of body composition could have been obtained in the present study by using the full scored patient-generated subjective global assessment instead of the PG-SGA SF. However, with outcome assessments performed at four different locations by different assessors, extensive training would have been required to ensure good inter-rater reliability in the subjective assessment [42]. The PG-SGA SF was chosen since it is easy to administrate, and it is designed to reflect 80–90% of the full scored patient-generated subjective global assessment score [42].

As results indicate, nutritional rehabilitation may still be indicated even though the weight is stable. According to the nutrition triage recommendations for the full scored patient-generated subjective global assessment, individuals with a score of 4–8 requires intervention by a dietitian in conjunction with a nurse or physician as indicated by symptoms, while a score ≥9 indicates a critical need for intervention. Since the PG-SGA SF is designed to reflect approximately 80–90% of the full scored patient-generated subjective global assessment score [42], the same cut-offs can potentially be used for the PG-SGA SF. Baseline data showed that 44% of participants in the intervention group had a PG-SGA SF score of 4–8 and 14% had a score of ≥9; hence, according to PG-SGA SF, 48% required intervention by a dietitian. In comparison, NRS 2002 only classified 11% as being at a nutritional risk. While the malnutrition score in NRS 2002 is primarily based on weight loss and decreased dietary intake, the PG-SGA SF furthermore includes information on nutrition impact symptoms and diet texture, and the different results obtained with the two tools may indicate that the nutritional rehabilitation needs in this population primarily concerns support to manage nutrition impact symptoms rather than support to gain weight. Hence, outcome measures of how individuals cope with nutrition impact symptoms may be more relevant.

Studies have shown that eating problems in HNC survivors frequently lead to social withdrawal [4,18,20], and in the pilot study of the intervention in the present trial, participants gained “Increased courage to eat” from the program [4]. Hence, the EORTC QLQ-H&N35 scale “Trouble with social eating” might have been a more relevant primary outcome. Significant improvement in this scale was seen in the intervention group, but with only 65 participants having complete data in this scale, the trial was not powered to show any difference between groups. Based on data from the present trial, 36 observations in each group would be required to detect a difference of 10 points (corresponding to a medium clinically relevant effect [57]) between groups with a power of 80% and a significance level of 5%. To detect a small effect (a difference between groups of 5 points), 141 participants would be required in each group.

Analyses of the effect on secondary outcomes indicated tendencies towards greater improvements in physical function in the intervention group, except for 30-s chair stand test, where a statistically significant improvement was seen in the control group. The same overall trend towards greater improvements in the intervention group was seen for several QOL scales and symptoms of anxiety, and the intervention group showed significant changes in approximately twice as many QOL scales than the control group. Notably, compared to the control group, participants in the intervention group had greater increases in the symptom scale “Felt ill”. While eight participants in the intervention group had an increase in symptom levels on this scale, the same was the case for one participant in the control group (data not shown). This scale is based on a single item and refers to whether the individual has been feeling ill during the preceding week. With the wording of the Danish EORTC QLQ-H&N35, feeling ill can either refer to acute illness or to a more generalized feeling of being categorized as suffering from a disease. Bearing in mind that most participants were 1–5 years posttreatment and their focus on the cancer and its late effects may have decreased over time, participating in an intensive rehabilitation program increases this focus again. This could potentially lead to an increased feeling of being affected by the cancer. This feeling is not uncommon in cancer rehabilitation which for some individuals comprise the acceptance of the long-term rehabilitation aim being to cope with the late effects rather than curing them [58].

The significant improvements seen in QOL scales in the pilot study of the intervention [4,24] were also seen in the intervention group of the present trial in addition to improvements in several other QOL scales. In their pilot study on the effect of a 1-week residential psychoeducational program at 4-week follow-up, Hammerlid et al. saw the greatest improvements in the EORTC QLQ-H&N37 scales, “Trouble eating” and “Problems enjoying your meals”, in HNC survivors 12–22 months posttreatment [19]. These scales do not directly translate to the current QLQ-H&N35 scales, but, consistent with our results, this could indicate that the residential programs support the HNC survivors’ ability to cope with physical symptoms rather than reducing physical symptom severity.

Another factor that could possibly have affected results of the present trial is the mode of participant recruitment and inclusion criteria. While recruitment in the pilot study was dependent on referral from physicians, inclusion in the present trial was based on self-reported interest in participation, and no further selection based on nutritional screening was performed. It can be hypothesized that some of the trial participants in the given study would have no measurable benefit of rehabilitation services no matter how effective the intervention, since they had relatively few or no late effects. Restricting inclusion to individuals who met certain criteria for nutritional status, nutritional risk, or presence of nutrition impact symptoms could potentially have led to other results. However, evidence is scarce on what these criteria optimally should be. Most nutrition screening tools validated in cancer patients are validated in the acute phase of the trajectory [42,59,60], and their applicability in HNC survivors >1-year posttreatment is unstudied. In the present trial, differences in baseline nutritional risk were seen when comparing NRS 2002 and PG-SGA SF. A secondary objective of the trial was in fact to test associations between participants’ development in outcome scores and their baseline scores in selected nutrition screening tools [24], and hence to assess the applicability of the different tools in this population. This secondary objective will be approached in future publications but could not have been pursued properly with further inclusion criteria in the trial. Inclusion through self-referral poses a risk that included participants are not necessarily the ones with greatest rehabilitation needs. This is already seen in existing rehabilitation services in Denmark even upon referral from health professionals [61,62]. Furthermore, being invited to participate in the rehabilitation program as a part of clinical study may have encouraged individuals to participate to support research despite few rehabilitation needs. However, compared to other NUTRI-HAB survey respondents, a greater proportion of trial participants (69% vs. 48%, *p* = 0.007, Appendix C, Table A4) had a REHPA scale score of 3 or above, which is REHPA’s inclusion criteria for their standard residential rehabilitation programs. The results of the present work support that access to rehabilitation services should be based on referral from health professionals rather than self-referral. Furthermore, results highlight the need for further explorative studies on relevant inclusion criteria for residential rehabilitation programs, not least because these are costly interventions. In the present study, EQ-5D-5L has been measured to allow for future economic evaluation of the program in different subgroups of participants.

Since the program required participants to be self-reliant and to participate actively, the most vulnerable HNC survivors may have been excluded. Recruitment through a nationwide survey gives a unique possibility to assess potential selection bias. Compared to the remaining survey population, more trial participants were female (*p* = 0.032), and more were diagnosed with pharyngeal cancer, while fewer were diagnosed with oral or laryngeal cancer (*p* = 0.011, Appendix C, Table A4). A great proportion of pharyngeal cancers are related to human papillomavirus, and these individuals tend to have a higher socioeconomic status and less alcohol and tobacco abuse [63]. Hence, their symptom level and rehabilitation needs may differ from oral or laryngeal cancer survivors’.

In addition to recruitment through a nationwide survey with data from a national clinical quality database, methodological strengths of the trial include randomization, blinded data analysis, and blinded interpretation of results. The use of a wait-list control group may have reduced participant drop-out, and with a relatively low attrition rate (8%) equally distributed across groups, the risk of attrition bias is considered low. Since concerns have been raised that wait-list control groups may overestimate intervention effect in randomized controlled trials [64], and the number of statistical tests in the present trial poses a risk of type 1 errors, no firm conclusions on the intervention effect on secondary outcomes can be drawn.

## 5. Conclusions

A multidisciplinary residential nutritional rehabilitation program had no effect on body weight in HNC survivors included based on self-reported interest in participation, but it may have effects on physical function and QOL. Since participants were 1–5 years posttreatment, nutritional rehabilitation needs primarily concerned support to manage nutrition impact symptoms rather than support to gain weight, and changes in body weight were not a relevant outcome for most participants. Consideration should be given to appropriate outcome measures in future clinical studies in this population. Further research on relevant inclusion criteria for referral to nutritional rehabilitation and the program’s effect in different subgroups of HNC survivors is needed.

## Figures and Tables

**Figure 1 nutrients-12-02117-f001:**
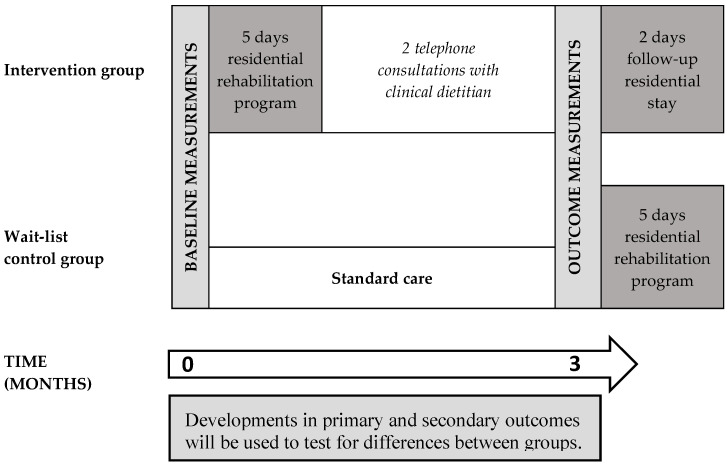
Timeline of the NUTRI-HAB trial from baseline to 3-month follow-up.

**Figure 2 nutrients-12-02117-f002:**
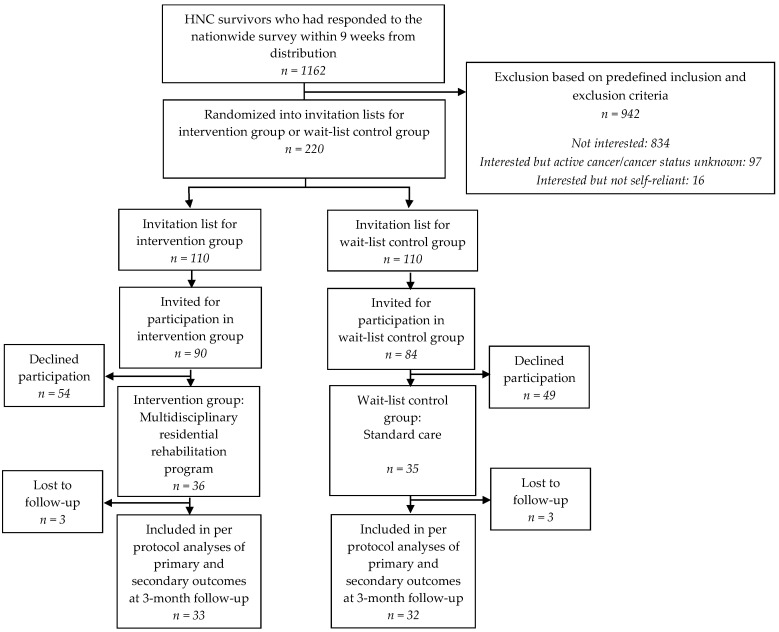
Flow chart of the NUTRI-HAB trial from baseline to 3-month follow-up.

**Table 1 nutrients-12-02117-t001:** Schedule for the initial five-day stay and two-day follow-up of the multidisciplinary residential nutritional rehabilitation program in the NUTRI-HAB trial.

Initial Five-Day Residential Stay					Follow-Up Residential Stay after 3 Months	
DAY 1	DAY 2	DAY 3	DAY 4	DAY 5	DAY 1	DAY 2
	Breakfast					Breakfast
	Morning assembly					Morning assembly
Arrival Welcome session and presentation of the program *(course leader and clinical dietitian)* Walk and talk	Practical kitchen workshop *(clinical dietitian)*	Psychological reactions to cancer *(psychologist)*	Physical activity *(physiotherapist)* Optional sessions: Fatigue and sleep problems *(nurse)* or Vocational counselling *(social worker)*	Motivation, goal setting, and action plans *(social worker and course leader)* Individual work and group discussion on action plans *(social worker and course leader)*	Arrival Welcome session and presentation of the program *(course leader and clinical dietitian)* What’s new within the last three months? *(course leader and clinical dietitian)*	Physical activity *(physiotherapist)* Optional sessions: Sexuality and intimacy *(sexologist)* or Meaning and values in life *(psychologist)*
Lunch					Lunch	
Introduction round *(course leader and central health professionals)* Theoretical session on management of eating problems *(clinical dietitian)*	Data collection: Physical tests and measurements *(physiotherapist)*	Swallowing exercises *(occupational therapist)* Individual dietary counselling *(clinical dietitian)*	Dental problems and oral hygiene *(dental hygienist)* Individual counselling *(depending on participant’s needs)* Massage therapy*(massage therapist)*	Closing session and farewell (*course leader and clinical dietitian*)	Data collection: Physical tests and measurements *(physiotherapist)*	Closing session and farewell *(course leader and clinical dietitian)*
	Yoga *(physiotherapist)*				Individual dietary counselling *(clinical dietitian)*	
Dinner					Dinner	
Social activity	Group conversation on existence *(priest)*					

**Table 2 nutrients-12-02117-t002:** Baseline characteristics of participants in the NUTRI-HAB trial.

	Intervention Group (*n* = 36)	Control Group (*n* = 35)
Age (years)	64.5 ± 6.7	64.0 ± 9.6
Gender		
Male	26 (72%)	20 (57%)
Female	10 (28%)	15 (43%)
Cancer diagnosis		
Larynx	6 (17%)	3 (9%)
Pharynx	30 (83%)	29 (83%)
Oral cavity	0	3 (9%)
Overall cancer stage		
I	6 (17%)	3 (9%)
II	7 (19%)	5 (14%)
III	5 (14%)	8 (23%)
IV	18 (50%)	19 (54%)
Tumour (T) stage		
T1	12 (33%)	8 (23%)
T2	9 (25%)	14 (40%)
T3	12 (33%)	9 (26%)
T4	3 (8%)	4 (11%)
Lymph node (N) stage		
N0	12 (33%)	8 (23%)
N1	4 (11%)	6 (17%)
N2	20 (56%)	21 (60%)
N3	0	0
Metastasis (M) stage		
M0	36 (100%)	35 (100%)
M1	0	0
Time interval from completion of radiation therapy		
12–23 months	13 (36%)	11 (31%)
24–35 months	6 (17%)	5 (14%)
36–47 months	7 (19%)	14 (40%)
48–59 months	10 (28%)	5 (14%)
Rehabilitation needs measured by the REHPA scale ^a,b^		
<3	13 (36%)	9 (26%)
≥3	23 (63%)	26 (74%)
Nutritional risk (NRS 2002)		
≥3 points	4 (11%)	2 (6%)
Nutritional risk and deficit (PG-SGA SF)		
4–8 points	16 (44%)	14 (40%)
≥9 points	5 (14%)	6 (17%)
BMI category		
Underweight (BMI < 18.5)	0	0
Normal weight (BMI 18.5–24.9)	17 (47%)	15 (43%)
Overweight (BMI 25.0–29.9)	13 (36%)	10 (29%)
Obese (BMI ≥ 30.0)	6 (17%)	10 (29%)
Current body weight vs. precancer body weight ^a^		
<95%	18 (53%)	17 (50%)
95–105%	14 (41%)	13 (38%)
>105%	2 (6%)	4 (12%)
Participant’s own evaluation of current body weight ^a^		
Too low	3 (8%)	5 (14%)
Appropriate	18 (50%)	12 (34%)
Too high	15 (42%)	18 (51%)

NRS 2002: Nutritional risk screening 2002, PG-SGA SF: The scored patient-generated subjective global assessment short form, BMI: Body mass index. Data are presented as means ± standard deviations or numbers and (percentages). The PG-SGA SF score can range from 0–36, and NRS 2002 score can range from 0–7. On both scales, a higher score indicates a greater nutritional risk. The REHPA scale ranges from ranges from 1–9, and a higher score indicates greater rehabilitation needs. ^a^ Self-reported data collected through the nationwide cross-sectional NUTRI-HAB survey prior to inclusion. ^b^ Used for stratification of randomisation.

**Table 3 nutrients-12-02117-t003:** Changes in physical measurements and tests from baseline to 3-month follow-up in the NUTRI-HAB trial. Changes from Baseline to 3-month Follow-up.

	Baseline Values		Changes from Baseline to 3-Month Follow-Up					
	Intervention Group	Control Group	Intervention Group	Control Group	Difference between Groups ^a^ *p*-Value	Effect Size Cohen’s d [95% Confidence Interval]	Adjusted Model ^b^	
							β	*p*-Value
Primary outcome								
Intention-to-treat analysis								
Body weight (kg) (36/35) ^c^	78.8 ± 2.3	79.3 ± 2.8	0.46 ± 0.43 ^d^	0.38 ± 0.56 ^d^	0.910		0.194	0.795
Per protocol analysis								
Body weight (kg) (29/30) ^c^	80.4 ± 12.8	77.8 ± 16.5	0.45 ± 1.66 ^d^	0.41 ± 3.06 ^d^	0.958	0.01 [−0.50, 0.52]	0.215	0.752
Secondary outcomes								
Physical measurements and tests								
Body mass index (kg/m^2^) (29/30) ^c^	26.7 ± 4.4	27.0 ± 5.1	0.45 ± 1.66 ^d^	0.41 ± 3.06 ^d^	0.958	0.01 [−0.50, 0.52]	0.215	0.752
Maximal mouth opening (mm) (29/29) ^c^	47.7 ± 7.1	42.8 ± 10.1	0.6 ± 1.6	−0.3 ± 2.1	0.088	0.46 [−0.07, 0.98]	0.962	0.072
Maximal hand grip strength (kg) (29/30) ^c^	39.4 ± 9.2	39.3 ± 13.0	1.3 ± 3.8	−0.6 ± 3.3	**0.038**	0.55 [0.03, 1.07]	1.950	**0.042**
30-s chair stand test (number of repetitions) (28/29) ^c^	15.1 ± 4.2	13.8 ± 4.1	0.5 ± 2.3	2.3 ± 3.1 *	**0.012**	−0.69 [−1.22, −0.15]	−2.074	**0.008**
6-min walk test (m) (28/28)^c^	562.7 ± 72.1	572.9 ± 115.7	34.6 ± 43.4 *^,#^	8.5 ± 57.7	0.061	0.51 [−0.02, 1.04]	18.620	0.192

Baseline values and changes within groups are shown as means ± standard deviations (standard error in intention-to-treat analysis). Significant *p*-values are highlighted in bold. ^a^ Differences between groups are tested with linear regression in intention-to-treat analysis and with a two-sample two-sided *t*-test in per protocol analyses. ^b^ Differences between groups assessed in a multiple linear regression model including gender, time interval (months) posttreatment, and rehabilitation needs assessed by the REHPA scale. ^c^ n included in analyses in (intervention/control) groups. ^d^ Changes in body weight and body mass index from baseline to 3-month follow-up is shown in percent. * Statistically significant change (*p* < 0.05) within group from baseline to 3-month follow-up tested with a paired two-sided *t*-test in per protocol analyses. Results are shown in Appendix B, Table A2. ^#^ Clinically relevant change within group from baseline to 3-month follow-up is defined as a difference between mean value at baseline and 3-month follow up of minimum 5% for weight [41], 5 kg for hand grip strength [54], and 14 m for 6-min walk test [55].

**Table 4 nutrients-12-02117-t004:** Changes in health-related quality of life and symptoms of anxiety and depression from baseline to 3-month follow-up in the NUTRI-HAB trial.

	Baseline Values		Changes from Baseline to 3-Month Follow-Up					
	Intervention Group	Control Group	Intervention Group	Control Group	Difference between Groups ^a^ *p*-Value	Effect Size (r)	Adjusted Model ^b^	
							β	*p*-Value
EQ-5D-5L								
VAS (32/32) ^c^	79.0 (52.0; 87.5)	75.0 (61.1; 85.5)	1.5 (−1.0; 10.0)	3.5 (−6.0;6.5)	0.672	0.05	2.319	0.523
Summary Index Score (32/32) ^c^	0.783 (0.719; 0.859)	0.787 (0.740; 0.847)	0.0 (−0.008; 0.043)	0.0 (−0.280; 0.034)	0.440	0.10	0.012	0.548
EORTC QLQ-C30								
Global health status/QOL (33/32) ^c^	66.7 (58.3; 83.3)	66.7 (54.2; 83.3)	0.0 (0; 16.7)	0.0 (0; 12.5)	0.870	−0.02	−0.310	0.943
Functional scales								
Physical functioning (33/32) ^c^	86.7 (80.0; 100)	93.3 (76.7; 100)	0.0 (0; 6.7)	0.0 (−6.7; 0)	0.070	0.22	4.622	0.102
Role functioning (33/32) ^c^	83.3 (66.7; 100)	83.3 (66.7; 100)	0.0 (0; 16.7)	0.0 (−16.7; 0)	**0.024**	0.28	9.630	**0.041**
Emotional functioning (33/32) ^c^	83.3 (66.7; 100)	83.3 (66.7; 95.8)	0.0 (0; 0)	0.0 (0; 8.3)	0.416	−0.10	−2.740	0.464
Cognitive functioning (33/32) ^c^	83.3 (50.0; 83.3)	83.3 (66.7; 91.7)	0.0 (0; 16.7) *	0.0 (0; 0)	0.100	0.20	5.756	0.088
Social functioning (33/32) ^c^	83.3 (66.7; 100)	100.0 (75.0; 100)	0.0 (0; 16.7)	0.0 (0; 0)	0.211	0.16	5.525	0.238
Symptom scales/items								
Fatigue (33/32) ^c^	33.3 (11.1; 44.4)	27.8 (11.1; 33.3)	0.0 (−11.1; 0)	0.0 (0; 11.1)	**0.050**	−0.24	−8.161	0.053
Nausea and vomiting (33/32) ^c^	0.0 (0; 0)	0.0 (0; 8.3)	0.0 (0; 0)	0.0 (−8.3; 0)	0.723	0.04	1.054	0.787
Pain (33/32) ^c^	16.7 (0; 33.3)	16.7 (0; 33.3)	0.0 (−16.7; 0)	0.0 (0; 16.7)	**0.029**	−0.27	−8.536	**0.048**
Dyspnoea (33/32) ^c^	0.0 (0; 33.3)	0.0 (0; 33.3)	0.0 (0; 0)	0.0 (0; 0)	0.978	−0.003	1.284	0.750
Insomnia (33/32) ^c^	33.3 (0; 33.3)	33.3 (0; 50.0)	0.0 (0; 0)	0.0 (0; 0)	0.856	0.02	0.663	0.907
Appetite loss (33/32) ^c^	0.0 (0; 33.3)	0.0 (0; 33.3)	0.0 (−33.3; 0)	0.0 (0; 0)	0.879	−0.02	−5.582	0.383
Constipation (33/32) ^c^	0.0 (0; 33.3)	0.0 (0; 33.3)	0.0 (0; 0)	0.0 (0; 0)	0.785	0.03	0.222	0.965
Diarrhoea (33/32) ^c^	0.0 (0; 0)	0.0 (0; 0)	0.0 (0; 0)	0.0 (0; 0)	0.776	0.04	0.297	0.943
Financial difficulties (33/32) ^c^	0.0 (0; 33.3)	0.0 (0; 33.3)	0.0 (0; 0)	0.0 (0; 0)	0.807	0.03	1.425	0.766
EORTC QLQ-H&N35								
Symptom scales/items								
Pain (33/32) ^c^	25.0 (8.3; 33.3)	16.7 (8.3; 37.5)	0.0 (−8.3; 0)	0.0 (−8.3; 8.3)	0.507	−0.08	−5.046	0.316
Swallowing (33/32) ^c^	16.7 (8.3; 33.3)	25.0 (12.5; 25.0)	0.0 (−8.3; 0) *	−8.3 (−12.5; 0) *	0.760	0.04	−1.409	0.691
Senses problems (33/32) ^c^	33.3 (16.7; 50.0)	25.0 (8.3; 66.7)	0.0 (−16.7; 0)	0.0 (−16.7; 0) *	0.592	0.07	3.336	0.366
Speech problems (33/32) ^c^	22.2 (11.1; 33.3)	11.1 (5.6; 22.2)	0.0 (−11.1; 0) *	0.0 (0; 0)	0.136	−0.18	−6.306	**0.040**
Trouble with social eating (33/32) ^c^	25.0 (0; 33.3)	16.7 (0; 33.3)	0.0 (−16.7; 0) *	0.0 (−8.3; 0)	0.276	−0.14	−6.188	0.110
Trouble with social contact (33/32) ^c^	0.0 (0; 20.0)	3.3 (0; 16.7)	0.0 (−6.7; 0)	0.0 (−6.7; 0)	0.764	−0.04	0.135	0.965
Less sexuality (31/31) ^c^	33.3 (0; 66.7)	33.3 (0; 66.7)	0.0 (−16.7; 16.7)	0.0 (−33.3; 0)	0.534	0.08	0.808	0.925
Teeth (33/32) ^c^	0.0 (0; 66.7)	16.7 (0; 33.3)	0.0 (−33.3; 0) *	0.0 (0; 0)	0.198	−0.16	−8.512	0.144
Opening mouth (33/32) ^c^	0.0 (0; 33.3)	0.0 (0; 33.3)	0.0 (−33.3; 0) *	0.0 (0; 0)	0.148	−0.18	−5.607	0.256
Dry mouth (33/32) ^c^	66.7 (33.3; 100)	66.7 (33.3; 100)	0.0 (−33.3; 0) *	0.0 (0; 0)	0.202	−0.16	−10.064	0.102
Sticky saliva (32/32) ^c^	33.3 (33.3; 66.7)	50.0 (33.3; 100)	0.0 (−16.7; 0)	0.0 (−33.3; 0)	0.629	0.06	4.182	0.521
Coughing (33/32) ^c^	33.3 (0; 33.3)	33.3 (33.3; 33.3)	0.0 (−33.3; 0)	0.0 (−33.3; 0) *	0.300	0.13	10.130	0.149
Felt ill (33/32) ^c^	0.0 (0; 33.3)	0.0 (0; 33.3)	0.0 (0; 0)	0.0 (0; 0)	**0.020**	0.29	10.395	**0.020**
Pain-killers (33/32) ^c^	0.0 (0; 100)	100 (0; 100)	0.0 (0; 0)	0.0 (0; 0)	0.755	0.04	1.515	0.887
Nutritional supplements (33/32) ^c^	0.0 (0; 0)	0.0 (0; 100)	0.0 (0; 0)	0.0 (0; 0) *	**0.013**	0.31	31.465	**0.005**
Feeding tube (33/32) ^c^	0.0 (0; 0)	0.0 (0; 0)	0.0 (0; 0)	0.0 (0; 0)	0.313	−0.13	−7.271	0.240
Weight loss (33/32) ^c^	0.0 (0; 0)	0.0 (0; 0)	0.0 (0; 0)	0.0 (0; 0)	0.443	−0.10	−9.273	0.462
Weight gain (33/32) ^c^	0.0 (0; 0)	0.0 (0; 100)	0.0 (0; 0)	0.0 (0; 0)	0.155	0.18	16.499	0.226
HADS								
Anxiety (32/32) ^c^	4.5 (2.0; 8.0)	4.0 (1.0; 8.5)	−1.0 (−2.0; 1.0)	0.0 (−1.0; 1.0)	0.094	−0.21	−1.230	0.061
Depression (32/32) ^c^	4.0 (1.0; 7.5)	5.0 (2.0; 8.5)	0.0 (−1.0; 0.5)	0.0 (−2.0; 1.0)	0.789	0.03	−0.228	0.694

EORTC: European Organization for Research and Treatment of Cancer, HADS: Hospital anxiety and depression Scale. Baseline values and changes within groups are shown as medians and quartiles (Q1; Q3). Significant *p*-values are highlighted in bold. The EQ-5D-5L VAS ranges from 0–100, and the summary index calculated based on Danish values ranges from −0.624 to 1.0. A higher score indicates better self-rated health. The EORTC QLQ-C30 and QLQ-H&N35 scales range from 0–100. A higher score indicates a higher response level. Thus, a high score for a functional scale or global QOL indicates a high level of functioning/QOL and a high score on a symptom scale indicates a high symptom level. The HADS subscales range from 0–21, and a higher score indicates a higher symptom level. ^a^ Differences between groups are tested with the Mann–Whitney U test. ^b^ Differences between groups assessed in a multiple linear regression model including gender, time interval (months) posttreatment, and rehabilitation needs assessed by the REHPA scale. ^c^ n included in analyses in (intervention/control) groups. * Statistically significant change (*p* < 0.05) within group from baseline to 3-month follow-up tested with the Wilcoxon signed-rank test. Results are shown in Appendix B, Table A2.

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
