# Peer review of "Effects of a Multidisciplinary Residential Nutritional Rehabilitation Program in Head and Neck Cancer Survivors—Results from the NUTRI-HAB Randomized Controlled Trial"

_nutrients, 2020, doi:10.3390/nu12072117_

Round 1

Reviewer 1 Report

The paper seems non suitable for the journal: in fact, the role of nutrition in the design of the study is  a bit confusiong and it should be improved concerning the inclusion/exclusion criteria. Even the nutritional rehabilitation it is not well described, and it has not clear  if it can affect some physical improvement. Certainly, it doesn't affect body weight, that represent the first endpoint of the study. The study should be interesting, but it lacks of specific information about the nutritional programme, omega-3 fatty acid content, proteins and vegetable intake, important in defining the quality of the diet and the potential effect on mental e physical status

Reviewer 2 Report

General comments

The authors are to be commended for testing a novel intervention intending to address unmet needs for head and neck cancer survivors. The manuscript is well-written, clearly outlining a complex, multidisciplinary intervention delivered via randomized controlled trial.

The selection of percentage weight change in body weight as the primary outcome is acknowledged by the authors as a limitation and addressed appropriately in the discussion.

  • Since nutritional status utilizing a validated assessment tool versus weight/percentage weight change may be more representative of chronic nutritional issues, could you please comment on the decision to use the PG-SGA short form versus also completing the global assessment of muscle/adipose/fluid status to assess nutritional status and therefore determine the degree of malnutrition, if present?

This is an interesting study, highlighting the importance of selecting outcomes that are of importance to patients.  Since improvements were seen in physical function – could you please comment on:

  • The details of the physiotherapy intervention eg was this just for the duration of the residential programme or were participants given an exercise program to continue at home?
  • Was consideration given to direct methods of muscle status evaluation eg muscle mass vs muscle strength and function?

Offering a residential program is a unique approach to addressing unmet needs for head and neck cancer survivors. Could the authors please comment on the cost of delivering such a program and whether an economic evaluation of program delivery is under consideration?

Specific comments

roduction

Line 51 – “curatively” may read better as “curative”

Materials and Methods

Line 98- 99 – “curatively intended radiation therapy” may read better as “radiation therapy of curative intent”

Line 202-203 – Where nutritional risk is discussed, please elaborate here on whether nutritional status was assessed, and if so, which tools were applied?

Discussion

Lines 428 – 441- this paragraph discusses the use of the PG-SGA SF, however, completing the full Scored-PG-SGA including the global evaluation of muscle/adipose/fluid status would have provided additional valuable insights and enabled the diagnosis of malnutrition in the population studied. It’s an important distinction to make and the discussion could be strengthened by expanding on this.

Conclusion

The authors have made key observations in the discussion:

“nutritional rehabilitation may still be indicated even though the weight is stable” and “nutritional rehabilitation needs in this population primarily concerns support to manage nutrition impact symptoms rather than support to gain weight.” It would be good to see some reference to this in the conclusion – given that one of the key learnings is that consideration should be given to appropriate endpoints for this type of study in this population.
